# The 4.2 ka event is not remarkable in the context of Holocene climate variability

Nicholas P. McKay [1] ✉, Darrell S. Kaufman [1], Stéphanie H. Arcusa[1,2], Hannah R. Kolus[1,3], David C. Edge [1], Michael P. Erb [1], Chris L. Hancock [1], Cody C. Routson [1], Maurycy Żarczyński [1,4], Leah P. Marshall [1], Georgia K. Roberts[1] & Frank Telles[1]

The "4.2 ka event" is a commonly described abrupt climate excursion that occurred about 4200 years ago. However, the extent to which this event is coherent across regional and larger scales is unclear. To objectively assess climate excursions in the Holocene we compile 1142 paleoclimate datasets that span all continents and oceans and include a wide variety of archive and proxy types. We analyze these data to determine the timing, significance and spatial imprint of climate excursions using an objective method that quantifies local, regional and global significance. Site-level excursions in temperature and hydroclimate are common throughout the Holocene, but significant global-scale excursions are rare. The most prominent excursion occurred 8200 years ago, when cold and dry conditions formed a large, significant excursion centered in the North Atlantic. We find additional significant excursions between 1600 and 1000 years ago, which agree with tree-ring data and annual-scale paleoclimate reconstructions, adding confidence and context to our findings. In contrast, although some datasets show significant climate excursions 4200 years ago, they do not occur in large, coherent spatial regions. Consequently, like most other periods in the Holocene, the "4.2 ka event" is not a globally significant climate excursion.

Future climate will respond to anthropogenic and natural climate forcings and internal variability on all timescales. Understanding and mitigating forced anthropogenic changes is fundamental for societies to plan and adapt for the future; however, investigating how natural variability affects the range of plausible future climate trajectories is also necessary. On multidecadal and longer timescales climate models underestimate natural variability at regional scales[1], suggesting that model projections, our best tool for preparing for climate change, underestimate the amplitude and duration of fluctuations at small spatial scales where climate action is most practical. Additionally, paleoclimate data indicate that abrupt and persistent climate changes are characteristic of the climate system, making the study of past abrupt climate changes fundamental to societal efforts to prepare for and adapt to climate change.

There are many types and definitions of abrupt climate change[2–4]. Here we focus on climate "excursions": short-term deviations in climate, at local to global spatial scales, where the climate variable of interest rapidly departs from the range of observed variability for a comparatively short period of time before returning to the prior state. Excursions occur in weather and climate at all timescales from days to millennia, with substantially varying impacts. The canonical example of long-term climate excursions occurred during glacial climate, known as Dansgaard-Oeschger (DO) events[5]. These regime shifts had dramatic local impacts, with changes in surface temperature of more

---

[1]Northern Arizona University, School of Earth and Sustainability, Flagstaff, AZ, USA. [2]Arizona State University, School of Complex Adaptive Systems, Tempe, AZ, USA. [3]Rhodium Group, Washington, DC, USA. [4]Department of Geomorphology and Quaternary Geology, Faculty of Oceanography and Geography, University of Gdansk, 80309 Gdansk, Poland. ✉e-mail: nicholas.mckay@nau.edu

than 10 °C in about 10 years in Greenland[6], driven by atmosphere, ocean, vegetation and ice-sheet dynamics with global impacts[7]. These changes persisted for several centuries before returning to prior conditions.

Climate excursions are also observed during the relatively climatically stable Holocene, the current interglacial period which began 11,700 years ago. Early observations of deep-sea sediment from the North Atlantic indicated a persistent 1500-year cycle of sea-ice-rafted debris related to solar variability[8]. These cycles loosely correspond to climate change events inferred from 50 globally distributed records reviewed by Mayewski et al.[9] and to a lesser degree with the six cold events inferred from 46 records summarized by Wanner et al.[10]. These Holocene climate excursions have smaller amplitudes than their glacial counterparts, but are likely more relevant for future change as they occurred with near-modern boundary conditions.

The Common Era (1 AD to the present) has several well-studied century-scale climate excursions that have similar amplitudes and durations to those earlier in the Holocene. These include the Little Ice Age and Medieval Climate Anomaly, which, due to denser data coverage, are better understood than preceding Holocene excursions. Multiple studies have demonstrated that these excursions are not globally coherent in time or space[11,12]. Despite this, they are significant century-scale excursions when averaged at regional or global scales.

The two most widely discussed climate excursions in the Holocene prior to the Common Era are both named for approximately when they occurred, the "8.2" and "4.2" ka events. The 8.2 ka event is a widely observed cold excursion centered in the northeastern North Atlantic[13] and oxygen isotope anomalies registered in speleothems across southeast Asia[14]. The 4.2 ka event has been described as a "mid/low latitude aridification event"[15] and a "global megadrought"[16] that is "recognized across global records as a temperature anomaly"[17]. It is likely, however, that the perceived climatological importance of this drought event has been enhanced by archeological evidence of concurrent civilization collapse (e.g.,[18,19]). Although there are a number of individual records indicating that droughts of various durations occurred at around this time[20–25], others indicate increased moisture[26–31] or report no sign of any climate event[32–34]. Studies seeking to characterize the 4.2 ka event have focused on sites where a change has been observed and discount those where the event is not found, potentially leading to increased visibility of confirmatory data[35]. This confusion arises, in part, from the lack of a systematic testing of the event period against background variability and assessing its spatial coherence and extent, as well as from inconsistent criteria for both the timing and climate changes associated with the event[30,36–38].

In 2018, the International Commission on Stratigraphy (ICS) ratified the 8.2 and 4.2 ka events as subepoch boundaries for the Holocene[39], with global stratopoints in the NGRIP ice core in Greenland, and in a speleothem in Mawmluh Cave in India, respectively. This ratification coincides with a substantial increase of publications describing the 4.2 ka event, or more commonly, interpreting how humans or the environment respond to the presumed abrupt global climate change at sites around the world. It was the topic of a special issue of the journal *Climate of the Past*, which comprises 21 papers published from 2018–2019, and our literature review (see below) found that 73% of publications that include the term "4.2 ka event" were published in the past 5 years.

Despite the ratification of the 4.2 ka event as a globally significant chronostratigraphic marker by the ICS, and its growing prominence in the literature, the climate changes that occurred 4200 years ago remain poorly understood. Previous efforts at synthesizing the spatial characteristics of the event have relied on inconsistently applied criteria for both the timing and climate changes associated with the event[30,36–38] resulting in spatial patterns that are difficult to assess for likely climate dynamics or forcings. Unlike the 8.2 ka event, the 4.2 ka event is not coincident with any large-scale forcing or change in boundary conditions that seem likely to have driven the excursion. In contrast with the 8.2 ka event, which has been readily recognized in a review of Holocene cold events[10], the 4.2 ka event is less prevalent in systematic reviews of Holocene climate. For example, the 8.2 ka event is found in a recent synthesis of 275 globally distributed speleothem records that span the Holocene[14] and a recent data assimilation-based reconstruction of Holocene temperature[40]. The 4.2 ka event is not found in either.

There is growing evidence that abrupt climate changes occurred at many sites at many intervals throughout the Holocene[41], and that evidence for the 4.2 ka event in the Mediterranean and southwest Asia may be weaker than previously thought[42,43], raising the question of whether the climate changes observed during the 4.2 ka event were remarkable in the context of Holocene climate variability. Here we addressed this question using several approaches. First, we conducted an extensive literature review and meta-analysis to characterize how the 4.2 ka event is described by the paleoclimate community. Second, we assembled thousands of paleoclimate datasets that are indicative of changes in Holocene temperature or hydroclimate. This represents the largest collection of standardized and annotated multiproxy Holocene paleoclimate data to date. Finally, we analyzed each time series for climate excursions using an excursion detection method that we modified from Morrill et al.[13] that quantifies parametric uncertainty and compares the observations to a robust null hypothesis.

## Results and Discussion

### Meta-analysis of the "4.2 ka event" in peer-reviewed literature

We reviewed the literature on the 4.2 ka event to characterize how this interval is described across different archive types and regions by searching for peer-reviewed articles that contain the term "4.2 ka event" using the Web of Science bibliographic database (see Methods). This helped to characterize the climate variables that have been used to document an event, its prominence in different regions, and identify the minimum resolution of proxy records that would likely capture the event, thereby narrowing the search criteria for paleoclimate datasets to include in this study.

The search returned 88 articles (Supplemental Table S1), of which, 73% were published during the past five years (2018 and after). To expand the collection, we also included an additional 21 articles which Railsback et al.[30] identified as including the 4.2 ka event that were not found in our literature search. Among the 109 articles, evidence of the 4.2 ka event was described by the original authors and we could visibly recognize changes in timeseries of most of the records (99/109 = 91%). However, the type and timing of abrupt change varied substantially between these studies. We found excursions to be the most common (48/109 = 44%), followed by shifts in the long-term mean (21/109 = 19%). The remaining studies report a wide variety of different types of change (Table S1), and include many that tacitly or explicitly assumed a climatic deterioration had occurred at 4.2 ka and then used their data to evaluate its effect on human or natural systems. The timing of the 4.2 ka excursions in these records also varied substantially, beginning as early as 4.6 ka (median = 4.2 ka) and ending as recently as 3.4 ka (median = 4.0 ka), with an average duration of 325 ± 170 (±1 SD) years (Fig. 1). These results informed our choice to focus on climate excursions, as well as the parameters used in our excursion detection approach (Methods).

Despite the variety of abrupt changes reported in our literature search, most of the articles (72/109 = 66%) did state whether the climate around 4.2 ka was either wetter or drier relative to before and after. We use these records and their interpretations to characterize the published proxy climate data for the 4.2 ka event (Fig. S1a). In addition, we show the globally distributed evidence for wetter or drier conditions around 4.2 ka as compiled for an additional 106 sites by two review papers: one by Marchant and Hooghiemstra[36] focused on environmental shifts in tropical Africa and South America ca 4.0 ka

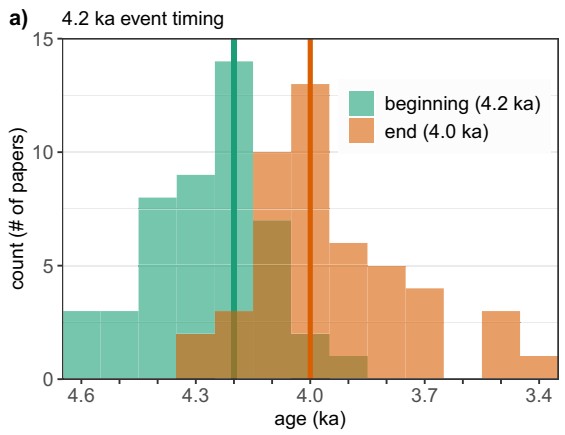
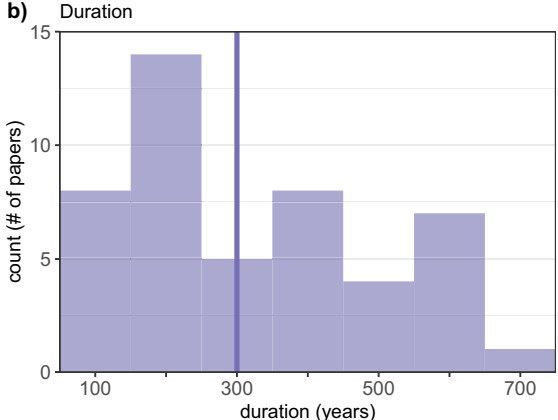

**Fig. 1 | The timing and duration of the 4.2 ka event in our literature review.** Timing and duration of 47 records with clear climate excursions from our literature review (Table S1). **a** Age of the beginning (green) and end (tan) of the event, with overlap between the two histograms shown as dark brown, and (**b**) duration of the event. Median values shown as bold vertical lines.

without any discussion of the 4.2 ka event, and a second by Wang et al.[37] that aimed to provide a global context for climatic changes as part of the 4.2 ka event. These two studies were later summarized by Renssen[38] to compare with model simulations of the 4.2 ka event. The approach used in these studies to document the abrupt climate changes around the 4.2 ka event is substantially different than the one featured in this study, as it focuses on longer-term (typically millennial-scale and longer) changes in the mean state, thereby affording a semi-independent counterpoint (Fig. S1b). Altogether, the complexity and variability of the type, magnitude and duration of the event complicates the interpretation of the synthesized results from the literature. Nevertheless, as our meta-analysis has shown, the description of the 4.2 ka event in the literature is variable and complex, and these maps (Fig. S1) reasonably characterize the spatial pattern of the event as interpreted by the authors in our literature review.

### Temperature and hydroclimate datasets

To investigate climate excursions during the 4.2 ka event and throughout the Holocene, we assembled a large compilation of paleoclimate data suitable for Holocene abrupt change studies (see Methods). We focused on compiling data suitable for identifying climate excursions, the type of abrupt change most commonly associated with the 4.2 ka event in our literature review and meta-analysis. Excursions in the literature were characterized as an event lasting from less than 100 to more than 500 years, with a median duration of ca. 300 years (Fig. 1). To capture excursions of this duration, we only considered intervals within time series where the mean temporal resolution is less than 100 years during the 400 years encompassing the event, as well as each of the 400-year intervals before and after the event window (Methods). Although the 4.2 ka event has been inferred from a wide variety of paleoenvironmental datasets, the majority of authors characterize the 4.2 ka event in terms of changes in hydroclimate, with some interpreted as temperature changes. Therefore, we also focus our analysis on these two climate variables and rely on the authors' interpretation of their proxy data. We compiled datasets (site-level data that often comprise multiple paleoclimatic timeseries) that met these criteria.

We found 896 datasets that have at least one timeseries interpreted as sensitive to hydroclimate, and 853 datasets with at least one timeseries interpreted to be sensitive to temperature (Fig. 2). Nearly all datasets are from different sites, but in some cases multiple datasets are included from the same site when produced by different studies, and often represent different proxies or methodologies. Many datasets ($n = 607$) also include both temperature

and hydroclimate-sensitive timeseries. Furthermore, just over half of the datasets ($n = 588$) also include multiple timeseries interpreted to record different seasonalities. Consequently, due to multiple proxies, seasonalities, and climate interpretations, many datasets include multiple climate-sensitive timeseries. To avoid missing any potential excursions, for most datasets we analyzed many temperature- and hydroclimate-sensitive timeseries in each dataset, only using the most significant excursion for each time period. This choice, along with other parametrization and analytical-design choices, were made to avoid false negative results and to maximize the number of possible excursion candidates. Although this could have resulted in over-identification of excursions, our robust null hypothesis testing protects against false positives overly influencing the result (Methods).

Altogether, the data span all seven continents and five oceans; however, there is a significant spatial bias in the datasets, as 82% of the datasets are north of 30° N latitude. Temporally, data coverage is strongest from about 4000 to 200 yr BP and decreases in the early Holocene. By 12,000 years ago both temperature and hydroclimate have about a third of their Holocene maximum data density, but in both cases nearly 300 datasets are still available. Data coverage remains strong as it approaches the present, but because our change detection methodology requires reference windows before and after the excursion, the window centered on 0.6 ka is the most recent we can analyze. Eleven archive types are represented in the compilation. Lake sediment (45%) and peat (22%) are the most common, with marine sediment being the third most prevalent in the temperature datasets, whereas speleothems are the third most abundant for hydroclimate datasets.

### Temperature and hydroclimate excursions in the Holocene

To objectively identify the occurrence of climate excursions throughout the Holocene, we applied the excursion detection methodology of Morrill et al.[13] after enhancing it by including parametric uncertainty and quantifying the significance of excursions relative to a robust null hypothesis (see Methods). We applied the excursion detector over the Holocene in 400 ±100-yr windows with sliding 200-yr steps for all relevant timeseries in all 1142 datasets in the compilation (Dataset S1 and Supplementary Bibliography). Figure 3 shows the spatially-weighted mean result of the excursion detection algorithm (i.e., the fraction of excursions found given parametric uncertainty) for 200-year steps intervals through the Holocene. These values range between 0.1 and 0.2 for both temperature and hydroclimate excursions through the Holocene. Because these values combine excursion

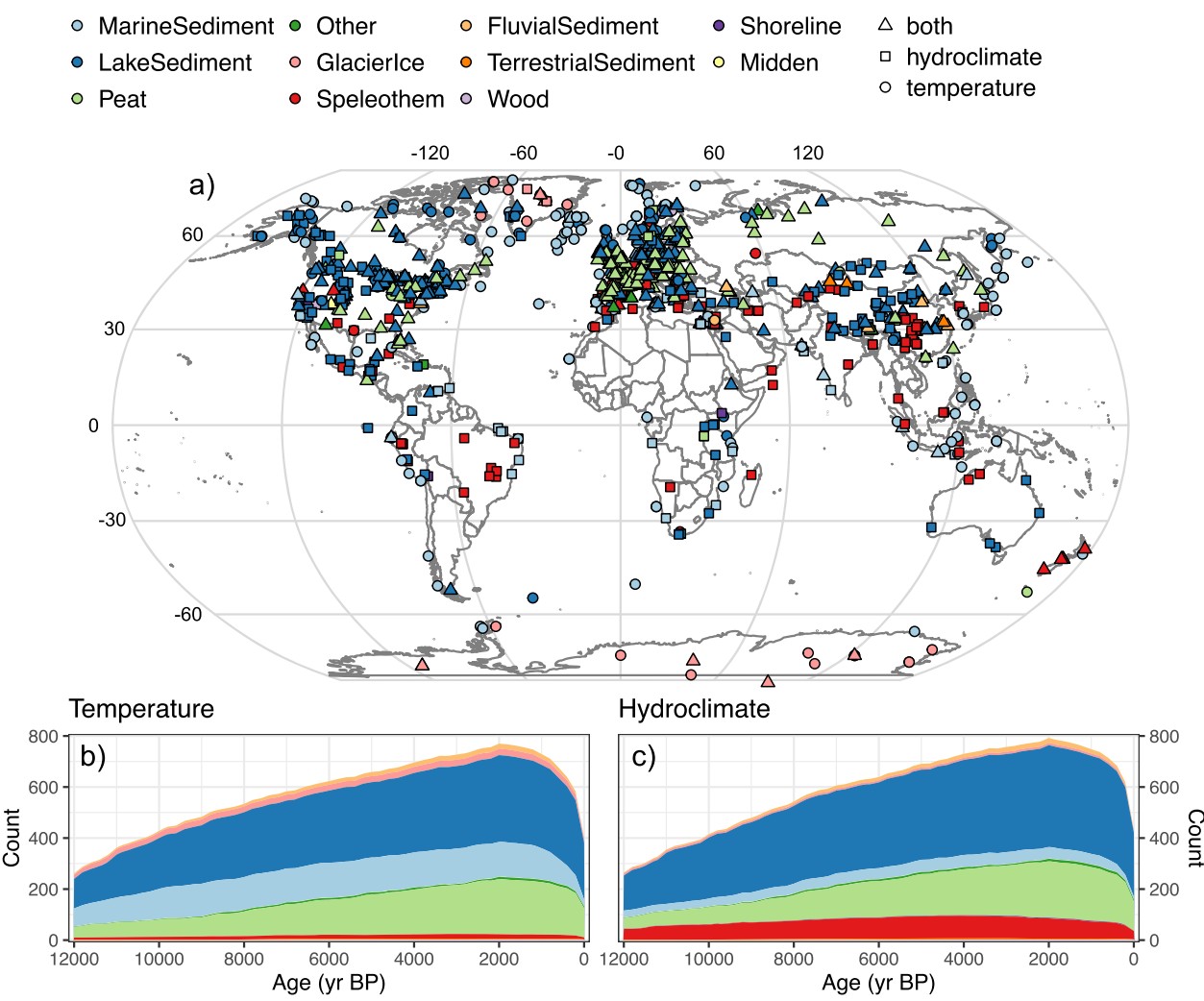

**Fig. 2 | Spatial and temporal coverage of analyzed paleoclimate data.** Location (**a**) and temporal availability of Holocene proxy records of (**b**) temperature and (**c**) hydroclimate analyzed in this study. In all panels colors correspond to archive type. Site details and references to original studies are listed in Supplementary Dataset S1.

magnitude, fraction of datasets with excursions and spatial weighting into a single metric, it can be hard to interpret. More simply, we find significant excursions in temperature in an average of 15% of datasets for any given $400 \pm 100$ yr interval, and significant hydroclimate excursions in 12%, although the fraction varies considerably between time periods (Fig. S2). These results differ markedly from our literature review, where of the 178 total moisture-sensitive records, 60% show evidence for drying. This marks an important distinction between our objective approach and our literature review. Our literature search targeted articles related to the 4.2 ka event or those that have been used to characterize the hydroclimate anomalies associated with the event, and rarely included studies for which evidence of the 4.2 ka event is lacking entirely. Taking an objective approach therefore appears to be critical to characterize the climatic significance of abrupt change events and to avoid confirmation bias.

For most of the Holocene, the occurrence of climate excursions is statistically insignificant (i.e., within the range expected relative to our null hypothesis), however several periods stand out as significant ($p < 0.05$), with synchronous excursions across many datasets. Notably, the proportion of time periods with significant excursions is generally in line with what would be expected from random chance (6.7% of the tested intervals exceed the 95% confidence bounds). Nevertheless, most of these intervals remain significant even when accounting for test-multiplicity using a Holm–Bonferroni correction[44]

modified to reflect the overlapping test windows and the corresponding reduction in degrees of freedom (Fig. 3).

Significant excursions are common at sites (83% of datasets show at least one significant excursion), and significant warm, cold, wet and dry excursions occurred at multiple sites for every interval we analyzed (Fig. S2). Together, this highlights that excursions are common features of Holocene paleoclimate records at the site level. Despite this, for most time periods, climate excursions are rarely spatially coherent at large scales, or concentrated into a single climatic variable or direction. We do, however, find several significant excursions even when aggregated across the whole of our dataset, in either temperature, hydroclimate or both throughout the Holocene. Globally (to the extent of our data coverage) significant warm excursions occurred ca. 11.4, 4.8, 2.6 and 1.0 ka, and significant cold excursions occurred ca. 8.2 and 1.6 ka (Fig. 3a). Significant excursions in hydroclimate were less common, with one significant wet excursion occurring at 4.8 ka, and the largest dry excursion at 8.2 ka (Fig. 3b). Although the dry excursion at 8.2 ka became insignificant following the test-multiplicity correction, it remains significant for both temperature and hydroclimate if we consider net excursions (wet/warm minus cold/dry), and is highly significant when we consider the joint occurrence of cold and dry excursions (Fig. 3c).

Two major previous syntheses of Holocene climate variability have emphasized the occurrence of cold and/or dry events[9,10], whereas

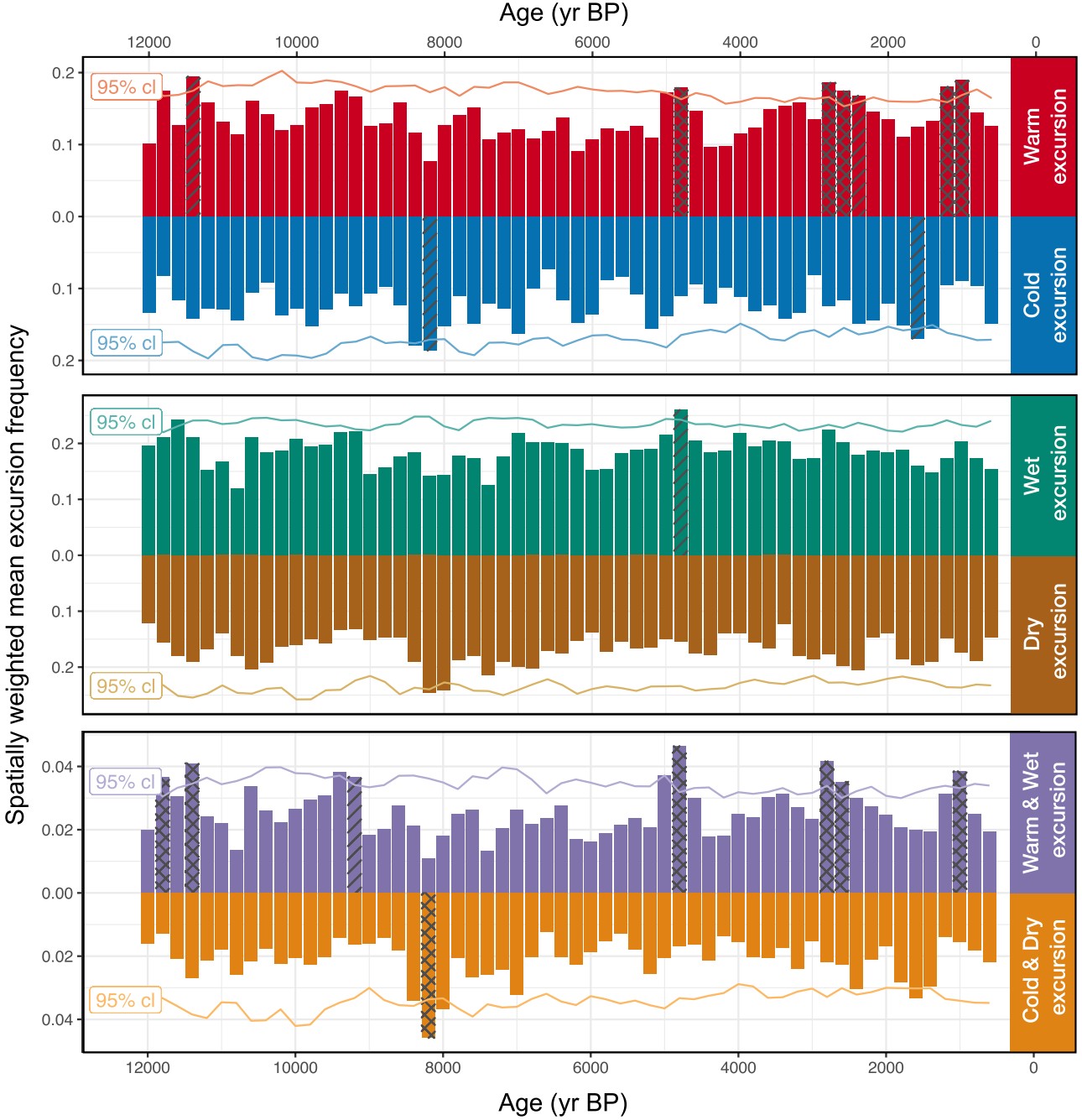

**Fig. 3 | Area-weighted average of site-level climate excursion frequency through the Holocene.** At each site, parametric uncertainty is considered, and the fraction of detected excursions for both directions in (**a**) temperature, and (**b**) hydroclimate range from 0 to 1. The occurrence and significance of (**c**) combined warm/wet and cold/dry excursions is shown as the product of wet/warm and cold/dry excursion frequency. Each bar shows the area- and uncertainty-weighted proportion of records with excursions in $400 \pm 100$ year windows. The axes for cold, dry, and cold/dry excursions are inverted. The excursion detection analysis is repeated in 200-year intervals across the Holocene. Colored lines show the 95% confidence interval (cl) based on null hypothesis testing. Due to test multiplicity however, we expect some apparently significant results to occur due to chance alone. Cross-hatched and single-hatched bars indicate intervals that remain significant at the 0.05 and 0.10 levels, respectively, after using a modified Holm–Bonferroni correction. See text for details.

we find warm events to be the most common type of significant climate excursion in the Holocene. Both of the significant cold excursions (8.2 and 1.6 ka) and the largest dry excursion (8.2 ka) identified in our analysis are consistent with previous syntheses, although the dry excursion is statistically insignificant following the test-multiplicity correction. We do not find evidence for additional significant cold or dry excursions reported in these syntheses. Century-scale cold excursions in the Common Era are often attributed to clusters of large volcanic eruptions[45,46], and a model simulation with realistic volcanic forcing over the past 8 kyr highlights the role of volcanism in cold excursions throughout the Holocene[47]. Cold excursions in our analysis often coincide with volcanically-driven increases in aerosol optical depth over the past 8 kyr, although only the excursion at 1.6 ka is significant in our analysis. Volcanism is a compelling mechanism to explain the observed cold excursions over the past 8 kyr. However, we find more significant warm than cold excursions in our analysis which

may be more difficult to link to external forcings, although the detected warm periods do tend to occur in the centuries following those with enhanced volcanic activity.

Interestingly, the most significant excursions in hydroclimate coincide with significant excursions in temperature, and in the direction predicted by the temperature-water vapor feedback. Every significant excursion in temperature becomes more significant when considering joint wet/warm or cold/dry excursions, with the exception of the cold excursion at 1.6 ka (Fig. 3c). Moreover, the spatial patterns of these temperature and hydroclimate excursions are often dynamically consistent, indicative of regional climate changes that are expressed across multiple aspects of the climate system. The cold and dry excursions centered on 8200 year BP (the 8.2 ka event) are well studied, as are Common Era temperature excursions centered at 1.6 and 1.0 ka. These time periods are discussed in detail below. However, the warm and wet excursions that occurred 4.8 ka are not widely discussed in the literature. Significant warm excursions at this time are focused in the Western Tropical Pacific, East Asia and Antarctica, with widespread significant wet excursions in Asia and the Northwest United States. Generally, these regional excursions in both temperature and hydroclimate are consistent with strengthened Walker Circulation[48], including evidence for cold excursions in the Eastern Pacific, although there are only a few sites in that region at that time.

## Common Era excursions

The two significant excursions during the Common Era deserve special attention considering the large number of paleoclimate studies that focus on this interval, including those based on tree-ring data. The significant cold excursion at 1.6 ka has been previously discussed as a Holocene-scale cold event by Wanner et al.[10] and is consistent with the "Dark Ages Cold Period" in the Northern Hemisphere[49] and in Europe[17].

Interestingly, this cold interval appears to predate the period of strongest volcanic activity and lowest solar irradiance of the first millennium, which coincides with the "Late Antique Little Ice Age" between 536 and 660 CE (ca 1.4 ka)[50], although we find marginally significant excursions during that interval as well. Following this pronounced cold excursion we find a significant warm excursion centered at 1.0 ka, consistent with the Medieval Climate Anomaly.

Like the 8.2 ka event, the detection of significant cold and warm excursions during these broadly defined periods helps to contextualize the resolution and fidelity of our analysis. It is well known that the coldest and warmest multi-decadal periods of the Common Era prior to industrialization occurred at different times in different places[11,12]. However, despite this regional heterogeneity and asynchronicity, our 200-year steps still capture multi-century cold and warm fluctuations prior to the Little Ice Age. This is encouraging, since the majority of datasets that have previously been used to characterize Common Era climate were excluded from our analysis due to the requirement that datasets span at least 2000 years. It is worth noting however, that the spatial patterns of these excursions are not entirely consistent with broader reconstructions of Common Era climate. For example, compare the temperature excursions at 1.6 and 1.0 ka (Fig. S3) with Neukom et al.'s[12] Fig. 3b,d. We cannot examine the most recent Common Era climate excursion (i.e. the Little Ice Age), because our excursion detection methodology requires reference windows that are 400 ± 100 years in duration before and after event windows.

## The 8.2 ka event

The 8.2 ka event is characterized by a marked cold and dry excursion centered over Greenland and the North Atlantic (Fig. 4). This pattern extends through northern Europe and southern Scandinavia, and continues to southeast Asia where significant dry precipitation

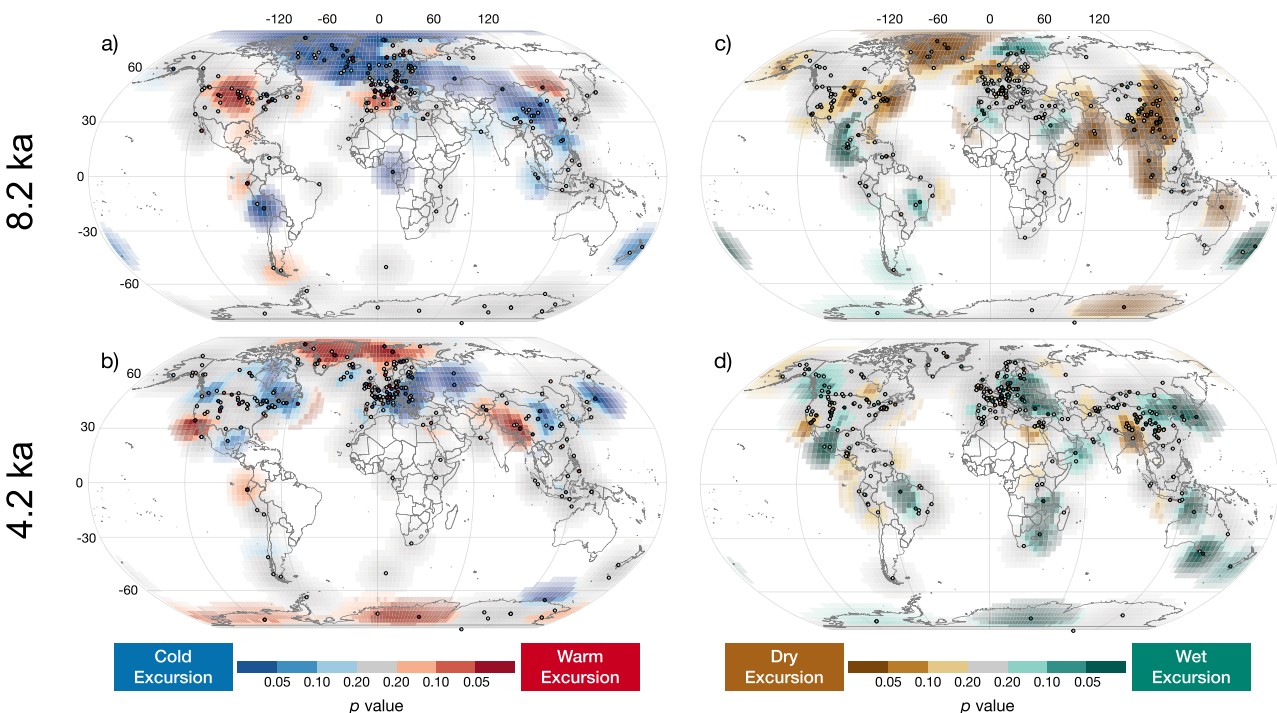

**Fig. 4 | Spatial expression of the 8.2 ka and 4.2 ka climate excursions.** Temperature (**a**, **b**) and moisture (**c**, **d**) excursions are shown for excursions centered at 8.2 ka (**a**, **c**) and 4.2 ka (**b**, **d**). The grid shading indicates the significance of regional net excursions. Darker red indicates more significant net temperature excursions (i.e., the number of warm excursions in a region significantly exceeds the number of cold excursions). Blue colors indicate regions where cold excursions significantly exceed the number of warm excursions. Green (wet) and brown (dry) colors express regional net excursions for hydroclimate. Transparency of grid shading increases with distance from the nearest site. Colored dots show the location of records with significant excursions at the 0.05 level, colored similarly to grid shading. White circles indicate no significant excursions at a site for the specified time period.

excursions are widespread. These patterns are generally consistent with previous syntheses[13,14]. The similarity of the spatial patterns of temperature and precipitation excursions produced by our approach for 8.2 ka event with those of previous studies is encouraging and suggests that our approach and dataset are well suited for detecting and quantifying the significance of excursions during the Holocene. Nevertheless, there are some key differences between our results and previous studies. Unlike Morrill et al.[13] we do not find evidence for warm excursions in the South Atlantic or Antarctica, or dry excursions in the Caribbean and northern South America. However, central North America and southern Europe show evidence for regional warm excursions. Some of these differences are due to differences in the collection of datasets, however adding parametric uncertainty and significance testing to the detection algorithm affected the results as well: some datasets that registered excursions in Morrill et al.[13] are not significant in our analysis.

In model simulations, the 8.2 ka event is typically forced by an influx of freshwater to the North Atlantic. This slows the thermohaline circulation, producing a cold anomaly in the North Atlantic[51,52] similar to the cold anomaly captured by our excursion detector. Temperature changes are less clear elsewhere, with some cooling of Northern Hemisphere continents and small or positive temperature changes in the Southern Hemisphere. Modeled precipitation changes are characterized by drier conditions in the Northern Hemisphere and wetter conditions in the Southern Hemisphere, with a southward shift of the Atlantic Inter-Tropical Convergence Zone (ITCZ)[51,52]. This agrees with some, but not all, of the detected proxy changes at that time (Fig. 4).

## The 4.2 ka event

Many sites ($n = 52$) record significant excursions in either temperature or hydroclimate during the 4.2 ka event. However, these represent a small fraction of the datasets analyzed (6.9% of hydroclimate datasets and 5.5% for temperature). For the most part, these excursions do not occur in large or coherent spatial regions. The largest region that shows a significant dry excursion is in southern Asia and includes Mawmluh Cave–the site used to define the 4.2 ka subepoch boundary for the Holocene[39]. Notably, we do not detect a significant excursion in the $\delta^{18}O$ record from Mawmluh Cave[53], because the large excursion at the end of that record is centered at 4.0 ka. Additional speleothems from Mawmluh Cave[54], which show notably different patterns ca. 4 ka than those of Berkelhammer et al.[53], were too short to be included in our analysis. Whereas differences in age models can affect our analysis, none of the 200-year periods near 4.2 ka stand out as exceptional, suggesting that age offsets are unlikely to explain the apparent lack of a significant multi-regional excursion during this interval (Fig. 3).

Much of the discussion of the 4.2 ka event has focused on megadrought in the Mediterranean and Middle East where excursions around 4200 years ago[20,22,30,55] are thought to have influenced ancient civilizations[16,18,19], despite the complexity in the evidence for the 4.2 ka event in this region[42,43]. We find little support for an arid excursion centered at 4.2 ka in the region (Fig. 4). The most notable regional excursion at 4.2 ka is a significant wet excursion spanning much of Europe. Previous syntheses have identified some evidence for anomalously wet conditions in the region ca. 4.2 ka[30,37], although it is not frequently discussed in the literature.

We find many more events in the region centered at 4.0 ka, along with a significant arid excursion throughout the broader eastern Mediterranean region (Fig. S3). Previous qualitative syntheses of the 4.2 ka event have often included excursions centered on 4.0 ka, and further away in time[30,55,56]. Although these events could be synchronous when considering the chronological uncertainty of their datasets, the prevalence of abrupt events throughout the Holocene means that one could readily use time uncertainty to similarly aggregate events at any point in the Holocene. In our analysis, the two intervals adjacent in time to the pronounced 8.2 ka event (8.0 and 8.4 ka) also show

increased occurrence of excursions, potentially due to chronological uncertainty (Fig. 3). The 4.2 ka event does not appear remarkable at hemispheric to global scales at 4.2 ka, or at any nearby time window, suggesting that age uncertainty is not the primary reason we don't identify a large-scale significant excursion at 4.2 ka. Many locations appear to have experienced abrupt climate excursions at this time; however these events seem typical of Holocene climate variability.

It is possible that regions experienced multiple excursions ca. 4.2 ka that may show up in both the 4.0 and 4.2 ka windows in our analysis. Indeed, multiple studies have identified multiple excursions in this interval (e.g., refs. 53,57,58). Multiple excursions can occur within a 400-year window, which would be obscured in our analysis. Nevertheless, the detected excursions at 4.2 (Fig. 3) and 4.0 ka (Fig. S3) share similarities with the pattern of abrupt change reported in the literature (Fig. S1). This is more clear if we examine the significance of excursions at each site centered at either 4.2 or 4.0 ka (Fig. S4), an approach similar to that in previous studies where events that fell loosely around 4.2 ka were assumed to occur simultaneously. When examining both time periods we find evidence for significant excursions in the eastern Mediterranean, the Middle East and south central Asia, and parts of eastern north America, as well as wet and cold excursions in parts of Europe, Eastern Asia and the Pacific Northwest of North America. However, despite the similarities, the excursion detector revealed a more heterogeneous pattern, both in space and in time. Although it may be possible given uncertainties in the datasets that these changes occurred synchronously 4.2 ka, it seems more likely that they occurred in different places at different times between 3.8 and 4.4 ka. Similar spatially and temporally variable changes in climate occurred frequently on these timescales throughout the Holocene.

Unlike the 8.2 ka event, identifying a mechanism for the 4.2 ka event has remained challenging for the community. Ning et al.[59] suggest that multi-century-scale fluctuations in Atlantic Meridional Overturning Circulation (AMOC) superposed on gradual cooling at high latitudes initiated Neoglaciation and triggered the changes at lower latitudes. Yan and Liu[60] argue that observed megadrought can be attributed largely to internal variability, with interactions between the North Atlantic Oscillation and the Atlantic Multidecadal Oscillation driving sustained drought in the northern Hemisphere. Rensseni[38] proposed that internal variability in tropical sea surface temperatures (SST), amplified by vegetation changes during the desertification of North Africa was the most likely cause of the 4.2 ka event. Most recently, Chen et al.[61] suggest that changes in ENSO likely contributed to observed changes ca. 4.2 ka.

Much of the variety in proposed causes of the 4.2 ka event is likely attributable to varying definitions of the phenomenon, ranging in timescales from decades to millennia, occurring in different locations, and in differing aspects of the climate systems. These studies also characterize the climate changes occurring at 4.2 ka as climatologically coherent changes occurring on spatial scales ranging from large regions[38] to hemispheric[60], which is inconsistent with our results. Our finding that climate excursions ca. 4.2 ka are comparable to those throughout the Holocene suggests that the difficulty in identifying a mechanism driving the event is likely because the changes that occurred at the interval were driven by a mix of long and short-term forcings, feedbacks and internal variability, which has been typical for the Holocene, rather than a pronounced forcing comparable to the 8.2 ka event.

Finally, it is important to recognize that although we find no evidence to support a large-scale excursion in climate that is significant in the context of Holocene climate variability, we do find many sites and some regions that experienced significant excursions in climate around this time. These changes in climate may still have been highly impactful, even if the occurrence of these events is typical in the context of the Holocene.

### The 4.2 ka event is not remarkable in the context of Holocene climate variability

Every multi-century interval in the Holocene was marked by significant excursions at multiple sites, and the interval centered on 4.2 ka is no exception. Unlike the 8.2 ka event, as well as more recent excursions during the Common Era, the excursions at 4.2 ka are not remarkable in the context of the Holocene. We find no evidence that 4.2 ka event is a global-scale, or even a particularly significant regional scale phenomenon. Therefore, the 4.2 ka event is ill-suited to serve as a chronostratigraphic marker or a Global Boundary Stratotype Section and Point (GSSP). Furthermore, the prevalence of regional climate excursions in the Holocene, and the lack of global-scale phenomena, suggest that climate excursions should not be used as Holocene GSSPs more generally.

The lack of evidence for pronounced and widespread climate excursion during the 4.2 ka event suggest that the growing prevalence of the event in the literature is more related to its reported association with the decline of ancient civilizations, and the intriguing, but unsupported, hypothesis that it is a major climate event for the Holocene. This latter claim appears to have contributed to a growing tendency we observed in our literature review, where authors sought to understand the expression or impact of the 4.2 ka event at their site, assuming that it was a globally pervasive event, even when no event was present, or where the observed climate impacts were minimal.

The prevalence of century-scale changes in the Holocene warrants further investigation, as it highlights the propensity of the climate system to undergo significant changes in climate, frequently in the absence of known forcings. Climate model projections will be more useful to plan and adapt for future change if they include the full range of natural climate variability at all timescales, in addition to the influence of anthropogenic forcings. Therefore, better understanding the timing, patterns and especially the causes of such events is of important societal relevance.

## Methods

### Meta-analysis

We searched the *Web of Science* bibliographic database in April 2023 for peer-reviewed articles with the words "4.2 ka event" or similar terms in any field (e.g., title, abstract, key words) and described how the resulting articles characterize the 4.2 ka event. The Web of Science is a bibliographic database used widely to conduct comprehensive literature searches across over 21,000 international journal titles. We compiled the timing, duration and climate interpretation of the 4.2 ka event as represented in papers that presented a clear excursion within a continuous, independently dated time series. For records that did not determine numerical ages for the event, such as discontinuous records (e.g., fluvial deposits) and those in which the 4.2 ka event is represented as a hiatus or other gap, we only compiled the climate interpretation of the 4.2 ka event (Table S1).

### Paleoclimate data compilation

We took several steps to compile an extensive collection of paleoclimate data appropriate for investigating abrupt change in the Holocene. To ensure a sufficiently broad collection, our initial screening criteria were intentionally broad and inclusive. To be included in the compilation, the datasets must:

1. be interpreted to record variability in climate in a peer-reviewed journal article,
2. have a median Holocene temporal resolution of 500 years or better, although only intervals with a mean resolution of about 100 years (i.e., <25% of the $400 \pm 100$ analysis window) or less were analyzed,
3. span at least 2000 years over the past 12,000 years

To find data that meet these criteria we queried recent paleoclimate data compilations[62–67], and searched the World Data Service for

Paleoclimatology (WDS-Paleo) and Pangaea for datasets associated with the 4.2 ka event, including both those found in our meta-analysis and records from previous compilations and investigations of the 4.2 ka event. Although we included every dataset that we found that met these criteria, the data compilation is not comprehensive, as not every dataset described in the literature is available at a publicly archived repository, and there may be others we overlooked.

All of these datasets were formatted in the Linked PaleoData framework[68] and were given standardized metadata that describe the datasets and their interpretation, including the climate interpretation and seasonality. Many datasets include multiple climate-sensitive timeseries, derived from different proxies, or interpreted to reflect different seasonalities. The data compilation is publicly available at http://lipdverse.org/HoloceneAbruptChange/current_version/.

### Abrupt change detection

Quantitatively identifying change points to detect regime shifts has received significant attention for climate time-series data[69–72], most of which are evenly distributed and absolutely dated. Alternative approaches have been developed for longer and unevenly spaced proxy climate records, however most approaches have focused on long-term changes in the mean[73,74] or the trend[75], rather than multi-century-scale excursions from the mean, which we identified from our meta-analysis as the most common characterization of the 4.2 ka event.

Multiple approaches have been developed to detect excursions as well. Recently, Parker and Harrison[14] identified excursions by using a change point detection as described above, focusing on periods when two significant break points occurred during a short interval (e.g., <300 years). Ön et al.[55] used Causal Impact Analysis in a Bayesian framework to detect excursions in a Bayesian structural time series approach quantifying the response to a theoretical excursion forcing. Other approaches have looked to identify excursions more directly, by quantifying departures from long-term trends[10,13,76,77].

One of these latter approaches, developed by Morrill et al.[13] identifies excursions by defining an event window along with two reference windows preceding and following the hypothesized event. An excursion is identified if two consecutive observations within the event window surpass the two standard deviation ranges of both reference windows (Fig. S5). This approach has the benefit that it directly tests the presence of excursions in paleoclimate data without any assumptions about the forcings, and has been successfully applied to Holocene paleoclimate data. However, the method is prone to parametric uncertainty, as the width of the detection or reference windows, the exceedance threshold and the required number of points above that threshold are somewhat arbitrary and make the results vulnerable to reasonable changes in these parameters.

### Excursion detection methodology

In this study, we expanded upon the approach of Morrill et al.[13] in three ways. First, we enhanced the detector's capacity to account for parametric uncertainty within the analysis. As with many change detection algorithms, this technique is highly parametric, making the binary outcome susceptible to various choices such as event and reference window durations, number of standard deviations for threshold exceedance, and consecutive points requirements for event registration. The improved algorithm considers a broad range of plausible parametric options, drawn from a distribution informed by the meta-analysis, producing an ensemble-based probability estimate rather than a pass-fail result.

Second, we introduced a robust null hypothesis methodology to assess result significance. Given the frequent autocorrelation and variations in temporal spacing within paleoclimate time series, random occurrences of excursions should be anticipated, differing in frequency across datasets. To address this, for each time series, we run

the excursion test on an ensemble (n = 100) of synthetic datasets, mimicking the real data's temporal spacing and spectral characteristics within the reference and event window range. This null hypothesis is analyzed using the same distribution of parametric choices as the real data. By contrasting the distribution of excursion fractions from null outcomes with those from actual data, we inferred the empirical *p*-value, quantifying the likelihood of the observed excursion arising by chance.

Third, we extend this robust null hypothesis approach to multiple datasets spatially. Applying the same approach, we investigate the likelihood that the observed frequency of excursions identified within a region happened by chance. We apply this approach using a global 3° × 3° grid. At each grid point, we identify all datasets within 1500 km, and calculate a weighted average of the probability of excursion over those datasets. These spatial scales and the weighting were chosen to balance the variable spatial density of the network and to generally align with spatiotemporal distance scales in the climate system[78]. To emphasize more local excursions, the weights are inversely proportional to distance from the grid point, calculated using a Gaspari-Cohn function[79] with a 1500-km cut-off radius. To assess the significance of these spatial excursions, this analysis is repeated with the synthetic data used in the null hypothesis. The observed weighted probability average is then compared to the distribution of synthetic weighted probability average to calculate the empirical p-value.

We repeated this approach across all datasets using sliding windows with 200-year steps and centers spanning 12,000 to 600 yr BP. For each sliding window, the excursion detector was run with a parametric ensemble. The excursion and reference windows were drawn from normal distribution with a mean of 400 years and a standard deviation of 100 years. This choice was informed by the meta-analysis, which indicated a mean excursion duration of 325 ± 170 years, and we chose a window length on the longer side of this range to consistently capture events of this duration. For each parametric ensemble member, the data are linearly detrended over the period that spans both reference windows and the event window. The threshold for excursion detection, in standard deviation units, was also drawn from a normal distribution with a mean of 2 and a standard deviation of 0.25. Excursions cannot be analyzed in every window for every dataset; to be considered for analysis, the event window and each reference window must include at least 4 data points, and the linearly detrended autocorrelation (AR1) of the full span (including both reference windows and the event window) for each window must be less than 0.98, a threshold chosen to exclude very highly autocorrelated timeseries which violate the assumptions of the methodology. In all of the cases we checked, the excluded timeseries were statistically interpolated data.

Many datasets in our analysis included multiple temperature- or hydroclimate-sensitive records at the same site, often reflecting multiple proxies or seasonal interpretations. To make the results comparable across sites, we restrict multi-site analyzes to only one result for each location, time period, and climate variable. Rather than choosing which timeseries to use a priori, and potentially missing excursions or mischaracterizing an event in a region, we took a conservative approach (likely to find more excursions) by calculating the likelihood of excursion for all timeseries, and using the most significant result for each site. This procedure overestimates the significance, but guards against false negative results.

To quantify the weighted fraction of excursions across the entire dataset (Fig. 3), we calculate the excursion fraction for each dataset, and then calculate a global weighted average to account for variable spatial density. The weights are calculated as 1/n, where n is the number of datasets within hexagonal equal area grid cells with 500 km spacing between adjacent grid centers[80]. This procedure is repeated with the synthetic data to calculate 95% confidence intervals for global weighted mean probability.

In addition to the spatial probability of excursions, we also consider the significance of the difference between the occurrence of positive and negative excursions (as in Fig. 4). To calculate these p-values we 1) subtract the weighted probability average of negative excursions from those of positive excursions, then 2) compare those values to the distribution of synthetic net excursions to estimate the likelihood of random occurrence. These enhancements to the Morrill et al.[13] excursion detector have been integrated into the publicly accessible Abrupt Change Toolkit in R[81].

All of the maps in this study (Figs. 2, 4, S1, S3, and S4) were created using R (v4.4.0)[82], relying heavily on the ggplot2[83], and sf[84] packages. Geographic and political boundaries were derived from the maps[85] (Figs. 2, 4, S3, and S4) and rworldmap[86] (Fig. S1) packages.

## Data availability

All of the data used in this study are publicly available at the World Data Service for Paleoclimatology, Pangaea or other long-term data repositories. Extended versions of the data, including additional metadata and formatting, are available as part of the Holocene Abrupt Change compilation on the LiPDverse (https://lipdverse.org). This study used version 0.11.0 of the compilation, which is available at http://lipdverse.org/HoloceneAbruptChange/0_11_0/, and future versions are available at http://lipdverse.org/HoloceneAbruptChange/current_version/.

## Code availability

The code used to conduct the analyzes in the study and produce the figures is available at[87]. This study relies heavily on the Abrupt Change Toolkit in R[81]. All of the datasets analyzed in this study, along with their references and the ages of significant events in each dataset are included in Dataset S1 and the Supplementary Bibliography.

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

## Acknowledgements
This study was originally conceived for the 4.2 ka BP Event International Workshop at the Università di Pisa, Italy in 2018. This work was funded by the Belmont Forum ACCEDE project (NSF-ICER-1929460) awarded to NPM and DSK. The Polish National Agency for Academic Exchange, project number BPN/BEK/2021/1/00133 funded MZ. This paper was greatly advanced as part of a seminar on Holocene Abrupt Change (EES-698) at Northern Arizona University in Spring 2023. Laura Larocca provided helpful feedback on an initial draft of this manuscript. Mindy Bell helped identify and document references for Dataset S1.

## Author contributions
N.P.M. and D.S.K. led and designed the study. D.S.K. led the meta-analysis, with contributions from D.C.E., M.P.E., C.L.H., C.C.R., M.Z., L.P.M., G.K.R., and F.T. C.C.R. and N.P.M. led the data synthesis and curation effort, with contributions from D.S.K., S.H.A., C.L.H., H.R.K., M.Z., L.P.M., G.K.R. and F.T. N.P.M. led the data analysis and visualization with contributions from S.H.A., D.C.E., M.P.E., C.L.H., H.R.K. and M.Z. N.P.M. wrote the manuscript, with major contributions from D.S.K., S.H.A. and H.R.K. N.P.M., D.S.K., S.H.A., H.R.K., D.C.E., M.P.E., C.L.H., C.C.R., M.Z., L.P.M., G.K.R., and F.T. reviewed, edited and commented on the manuscript.

## Competing interests
The authors declare no competing interests.
