## [Peer Review File · Nature Communications]

The 4.2 ka event is not remarkable in the context of
Holocene climate variabilityREVIEWER COMMENTS

Reviewer #1 (Remarks to the Author):

This is a timely, informative manuscript which address an important question relating to climate variability – what is the frequency of globally significant climate excursions in the Holocene. The manuscript concludes that centennial scale climate excursions are a pervasive feature of Holocene climate variability. However, in terms of global significance, while the 8.2ka event does pass the bar of being globally significant, the 4.2 ka event does not.

The project has a clearly defined focus and covers both a qualitative and quantitative assessment of the literature, ably and succinctly contrasting the two. The methodology appears robust and very thorough - it is discipline leading in this respect. The conclusions reached are broadly consistent with the data presented. The collated dataset presented is superb. However, I have a concern over the ease at which an excursion could be detected, which I think is susceptible to false positives.

The manuscript is well written for a broad audience not just of paleoclimatologists but also those who's fields are impacted by paleoclimate but who are not experts (archaeologists etc). The paper is therefore a good fit for Nature Communications. However, I worry that in some sections this simplification to a general audience goes too far, and a level of nuance is missing, such that the manuscript falls into a few traps that a prevalent in the literature: an expectation of homogenous climate responses, a false dichotomy of the impact of the 4.2 ka event, and lumping of temporally adjacent events. I'm reasonably confident the authors don't subscribe to these fallacies. But the manuscript does not convince me. These manuscript comments can be read as suggestions for improvement of the manuscript and the state of discourse in the literature.

Comments:

I worry the net has been cast too wide for excursions and that the risk of false positives is too high. I understand that the purpose of this study is to gather a large number of available records to maximise spatial coverage and therefore detect spatial patterns, but this

shouldn't necessarily come at the expense of data quality and reliable excursions. Other statistical assessments of the 4.2 ka event have used 20 and 15 year windows albeit with limited geographical scope, so while a larger window size is welcome to broaden the potential pool of records, there are compromises. Individually the combination of a 400-year window with 200 year steps, and only two datapoints needed for an excursion make me cautious. Together they ring alarm bells as being very susceptible to false positives. I hope that this caution on my part is due to poor phrasing, but I feel the need to raise a flag here in case it isn't.

400-year excursion detection intervals: The paper gives the impression that excursions are detected within 200 year intervals (line 201-202, 208-209. But the event detection window is 400+-100years long, with the analysis centered every 200 years (lines 511-512, and confirmed by Fig S5). Therefore an excursion does not have to occur within the 200 year interval depicted in figure 3. I understand the desire to capture both ends of the excursion, but if the excursion is big enough, having some of it hang into the reference window might not be a problem.

Describing a sliding window by its step frequency rather than its window size is not typical. I think this is potentially highly misleading. At a minimum the 400 year window size needs to alongside the 200 year step size somewhere in lines 198-217.

Two datapoints to characterise a 300 year long event. I have discomfort over the idea of 2 datapoints being sufficient to detect an excursion, even when autocorrelation is included in the significance testing. For high resolution records this could lead to very small short anomalies being included. For low resolution records a single outlier could be detected if there was an analytical rednoise (This often happens with outliers during laser ablation methods), or aliasing could occur for two consecutive points. Are the authors able to reassure me here?

Note, at first read, the manuscript states that the median resolution of the quantitative methodology is 500 years (line 438) versus 100 years for the qualitative methodology (Line

168). It was only by reading the code that I think a 100 ish (actually $\frac{1}{4}$ of the excursion window) resolution is in the quantitative section too (n_points_per_period right?)

Combining wet and dry excursions

I don't expect any centennial scale Holocene climate events to involve a change in significant planetary temperature, bar possibly the 8.2 ka event. And therefore by virtue of Clausius Clapeyron, a similar precipitation response. Therefore most, if not all, Holocene climate excursions will involve both warming and cooling, and wetting and drying in different places. The spatial analysis section of manuscript demonstrates this. Yet the time-series analysis separates the two (Figure 3), despite the authors commenting that combined wet and dry excursions provide for more significant climate excursions than the two alone for the 8.2ka excursion (line 236, line 257) and can be used to help characterise climate mechanisms related to the 4.8 ka excursion (line 278). I recommend up to 3 additional panels for figure 3 which combine wet and dry, hot and cold, and possibly all four would be useful to demonstrate this.

4.2, 4.0 and the recognition of contemporaneous events

The authors imply that Marchant & Hooghiemstra (2004) is an example of a compilation with poorer chronologies being tied to the 4.2ka event. Marchant & Hooghiemstra do not attempt to link their compilation to the 4.2ka event. It is only the work of others later who have seized upon proximal changes and assigned 4.2 to these changes. I think not recognising this does a disservice to Marchant & Hooghiemstra and should be recognised in the text (paragraph lines: 146-169), otherwise a straw man is created.

This raises the issue of multiple climate excursions occurring with a few hundred years of each other – which must surely happen, and their potential 'lumping' by aggregators, which the authors allude to on line 354-357. The 4.0 and 4.4ka events can be different to the 4.2 ka event, and have been shown to be in the literature. Leipe et al., (Quaternary International, 2014) found both 4.4 and 4.0 events in pollen records from Tso Moriri. Scropton et al. (2023, QSR 107837) found both 4.2 and 4.0 in different principal components

of an Indian Ocean regional synthesis using a quantitatively determined dataset. Even the GSSP stalagmite for the Meghalayan, KM-A has two positive isotopic shifts at 4.3 and 4.1 ka (Berkelhammer et al., 2012).

The 4.0 tropical climate shift can be its own climate event (though not an excursion by the nomenclature used in this study), even without global signature or universal regional signature. The 200 year window used in this study may not be able to distinguish these events, whereas the 15 and 20 year windows of Scropton et al. can. The paragraph between lines 351 and 362 could be updated to reflect separation of these two events in the literature, and the resolution limitations of a 400 year window.

Globally Significant vs Globally Impacting Event

I think the manuscript also conflates globally impacting vs globally significant events. We should expect smaller climate excursions to decay with distance from their focus regions. Thus what is a significant event in one region may be comparable to other centennial scale variability in others, and an undetectable part of noise in others. Identification of a climate event in a particular area is important to characterise the spatial extent of a climate events, regardless of whether it has a magnitude greater than other centennial scale variability. The idea of excursions “smaller than those observed throughout the record” (line 412) is not a problem if you’re trying to determine the global impact of an event. To truly understand the 4.2 kyr event we need to know where the event impacted, as well as where it was significantly different to other climate variability. The methodology used here addresses the latter. I think the authors should explicitly recognise that their analysis shows a limited spatial significance of the 4.2 ka event, not a limited spatial impact of the 4.2 ka event.

Minor Comments & Corrections

Line 35: “mitigating forced anthropogenic changes”

Line 93-95: This sentence needs rewriting to correct grammar. I doubt that your understanding has been ratified by the ICS.

Line 100: Sentence beginning “The 4.2 ka event” feels out of place. Could it be located either in the paragraph above, or below the 8.2 summary towards the bottom of the

paragraph.

Paragraph Line 126: This paragraph is a little hard to follow. I feel line 129 to 135 isn't necessary given what follows from Line 135 onwards. Key words and phrases from this section could be folded into what follows without losing any understanding.

Line 184: Swap order? "only using the most significant excursion at each site for each time period".

Fig 2b, c: y-axis is number not age?

Line 555: hyperlink is incorrect, link as spelt is correct

Fig S5: "four consecutive observations"?

Reviewer #1 (Remarks on code availability):

I have lightly assessed the code when checking other issues that came up during peer review. While I cannot vouch for its reusability, it does seem complete.

Reviewer #2 (Remarks to the Author):

This is a valuable addition to the literature and I recommend publication. It addresses a topic that pervades the literature—were there widespread globally significant changes in climate in the past? This topic has many adherents, and the literature is awash with claims of abrupt events in the Holocene, many of which lack scientific rigour and statistical significance. This paper uses a very large set of data and uses a rigorous statistical framework to address the topic. The results are particularly noteworthy in light of the International Geological Commission's decision to adopt the "Northgrippian" and "Meghalayan" (8.2ka and 4.2ka BP) as times of significant change in the Holocene. This paper affirms that the 8.2ka event was widespread, but clearly shows that there is no significance for the 4.2ka "event". It is thus an important rejoinder to the erroneous decision of the IGC. The authors do mention this point, but they could make more of it, I think. The IGC decision was simply wrong and should be reversed,

The paper clearly lays out its methods and data sources, allowing others to re-visit the topic if they wish. They also make the important point that, while there were certainly local or regional scale anomalies in the past, there were very few time periods when anomalies

extended to a global or even hemispheric scale.

A recent paper by Ön, Z. Bora (in QSR, 2023) is not cited and is relevant as it conducts a similar statistical assessment of abrupt changes in climate in the Mediterranean/Middle East region, where the original idea of a “4.2ka event” originated (he also found little evidence for it).

I think papers by Kathayat should be cited when discussing Mawmluh cave, as she collected more speleothems from the cave and found no support for Berkelhammer’s claim for a 4.2ka event, (which foolishly became the type section designated by the IGC).

Reviewer #3 (Remarks to the Author):

Dear Editor

I have now completed my review of the McKay et al. manuscript. I found it very clear and well written and the treatment of the data robust (even if I’m not an expert of some methods). Therefore, conclusions appear supported by this extensive review. I’m therefore completely in favor of the final acceptance of this manuscript. This represent an important contribution in the discussion on Holocene climate variability and its “formal” division. The former is probably more important than the latter.

I have some general considerations/comments and some minor point along the text

- 1) I think there is not enough discussion on the fact that age models can contain a very large error and there would be a potential impact on the conclusions. So, records are selected also from this point of view or not? I think it would be a monumental effort to obtain all these data and treat them consequently, but maybe a figure showing the range of the error associated to the age models could be a significant step in the quality of the manuscript. I can agree that the large number of data used may reduce the impact.
- 2) It is obviously not fundamental, but date are mostly concentrated in the Northern Hemisphere (as noted in the manuscript), Can it bias some conclusions?
- 3) It is clearly stated in the text that the interpretation of the proxies is based on the proposed interpretation in the original paper, it is also indicated that some proxies can be

seasonally biased. However, it is not well specified as this has been considered.

4) At the end seems that the 4.2 event is just one among many others during the Holocene, which has not enough spatial coherency to be considered a reliable boundary for the Middle to Late Holocene (even if the main conclusions are probably related to the meaning of climatic excursions during the Holocene). From the proposed analyses no other excursions seem to be suitable. Surprisingly (but not necessarily surprisingly) at 4.8 ka seems to occur some. Is it enough to say, searching for the 4.8 ka event? I think the author should try to suggest some line of research after this robust analyses. Or the main conclusion is: The Holocene has so very large temporal and regional variability that the boundary can be just an agreement?

Minor points

Why do you use the term “Medieval Warm Period” instead of “Medieval Climate Anomaly”? Is this because in your analyses “Medieval Period” is characterized by a clear temperature anomaly?

Lines 65-67 May be is a matter of room. But here some references would be necessary.

Figure 1: Fig. 1a, not clear the caption. There are three color in the figure, but it is unclear what it is the dark brown.

Figure 2 Maybe the caption of the figure 2 can be improved, for instance the color in figure b & c could not be obvious without an explanation.

Lines 259-261 I'm wondering if it is just related to the fact that many authors searched for cold and dry climate.....

Lines 343-346 here it would be correct to quote Bini et al. 2019. Despite the analyses of Bini et al was very qualitative, they show the very complex nature of the signal at regional scale.

Lines 351-354 Hmm....be honest this is not a trivial point. Of course this can be the same for many other “events” in the Holocene. But if we align all the record centered at 4.2 ka, is this

event become significant? I mean, the discussion of the error is not trivial. The authors give an answer in the following sentences but probably not necessarily all readers would be convinced.

When I was younger in the region where I worked there were very good evidences that the 4.2 event was dry and well chronologically constrained. Moving around I have discovered that it is not the case. I think part is related to the ability of the archives to record events, partly depending on the proxy you use.....but at the end McKay et al can have right and 4.2 event is not so prominent. One among other complex climatic configuration during the Holocene. So, my indication is that this manuscript can be accepted after minor/moderate modifications. Probably some more suggestions on future work need to be inserted in the conclusions and in particular the main effort in disentangling this complex climatic variability, since model, at best, can give us just a simplified vision of the future.

Reviewer #3 (Remarks on code availability):

ok

Response to reviewer comments.

We thank the reviewers for their thoughtful and constructive reviews. Please find our responses and comments on specific sections below. Reviewer comments are in black, and our responses are in blue.

Reviewer #1 (Remarks to the Author):

This is a timely, informative manuscript which address an important question relating to climate variability – what is the frequency of globally significant climate excursions in the Holocene. The manuscript concludes that centennial scale climate excursions are a pervasive feature of Holocene climate variability. However, in terms of global significance, while the 8.2ka event does pass the bar of being globally significant, the 4.2 ka event does not.

The project has a clearly defined focus and covers both a qualitative and quantitative assessment of the literature, ably and succinctly contrasting the two. The methodology appears robust and very thorough - it is discipline leading in this respect. The conclusions reached are broadly consistent with the data presented. The collated dataset presented is superb. However, I have a concern over the ease at which an excursion could be detected, which I think is susceptible to false positives.

The manuscript is well written for a broad audience not just of paleoclimatologists but also those who's fields are impacted by paleoclimate but who are not experts (archaeologists etc). The paper is therefore a good fit for Nature Communications. However, I worry that in some sections this simplification to a general audience goes too far, and a level of nuance is missing, such that the manuscript falls into a few traps that a prevalent in the literature: an expectation of homogenous climate responses, a false dichotomy of the impact of the 4.2 ka event, and lumping of temporally adjacent events. I'm reasonably confident the authors don't subscribe to these fallacies. But the manuscript does not convince me. These manuscript comments can be read as suggestions for improvement of the manuscript and the state of discourse in the literature.

We thank the reviewer for their positive and particularly constructive and helpful review. We respond to each of the comments, suggestions and other points below.

Comments:

I worry the net has been cast too wide for excursions and that the risk of false positives is too high. I understand that the purpose of this study is to gather a large number of available records to maximise spatial coverage and therefore detect spatial patterns, but this shouldn't necessarily come at the expense of data quality and reliable excursions. Other statistical assessments of the 4.2 ka event have used 20 and 15 year windows albeit with limited geographical scope, so while a larger window size is welcome to broaden the potential pool of records, there are compromises. Individually the combination of a 400-year window with 200 year steps, and only

two datapoints needed for an excursion make me cautious. Together they ring alarm bells as being very susceptible to false positives. I hope that this caution on my part is due to poor phrasing, but I feel the need to raise a flag here in case it isn't.

400-year excursion detection intervals: The paper gives the impression that excursions are detected within 200 year intervals (line 201-202, 208-209. But the event detection window is 400+-100years long, with the analysis centered every 200 years (lines 511-512, and confirmed by Fig S5). Therefore an excursion does not have to occur within the 200 year interval depicted in figure 3. I understand the desire to capture both ends of the excursion, but if the excursion is big enough, having some of it hang into the reference window might not be a problem.

We agree that we need to better clarify both the details of 400-year sliding windows, and the justification for the relatively broad windows, and the choice to only require two consecutive data points to qualify as an excursion. First, it's worth saying here (and has been added to the Results {L195-198} as well), that we intentionally chose loose criteria to include as many records that have been identified in the literature as 4.2 ka event excursions as possible - so we intentionally erred on the side having too many false positives, to avoid false negatives. That said, we did also test a wide range of configurations and parameters to find the options that most reasonably captured excursions like those identified in the literature. A relatively broad window turns out to be important to capture multi-century excursions, and can still capture shorter events that occur within the window. Whereas narrower window choices excluded some of the longer excursions identified in the literature, and although many authors have described the 4.2 ka event as a sub-centennial scale excursion, longer durations are also common in the literature (Figure 1). Our testing suggested that 400 ± 100 year windows were the best compromise to accommodate longer excursions while still being short enough to capture short excursions in higher resolution datasets.

Describing a sliding window by its step frequency rather than its window size is not typical. I think this is potentially highly misleading. At a minimum the 400 year window size needs to alongside the 200 year step size somewhere in lines 198-217.

Agreed, the phrasing here was misleading and we have revised and clarified this description.

Two datapoints to characterise a 300 year long event. I have discomfort over the idea of 2 datapoints being sufficient to detect an excursion, even when autocorrelation is included in the significance testing. For high resolution records this could lead to very small short anomalies being included. For low resolution records a single outlier could be detected if there was an analytical rednoise (This often happens with outliers during laser ablation methods), or aliasing could occur for two consecutive points. Are the authors able to reassure me here?

We agree that 2 data points can often feel insufficient to characterize a multi-century excursion, however this choice was made, again to error on the side of inclusion, and to minimize false negatives (relative to records identified in the literature at least). We did also explore modifications to this parameter, including some that scaled with resolution. Ultimately, there were no perfect solutions, and we decided to keep the parameter choice that Morrill et al., 2013 used.

Critically, however, we found that our null hypothesis testing procedure was critical for all of our parameter choices. Change detection techniques are often very sensitive to parameter choices, and this excursion detection methodology is no exception. Replicating the test across an ensemble of synthetic data that mimics the resolution, spacing, and autocorrelation structure of the dataset and is our primary and best guard against false positives.

Note, at first read, the manuscript states that the median resolution of the quantitative methodology is 500 years (line 438) versus 100 years for the qualitative methodology (Line 168). It was only by reading the code that I think a 100 ish (actually $\frac{1}{4}$ of the excursion window) resolution is in the quantitative section too (n_points_per_period right?)

That's correct, and we agree that this is confusing in the manuscript. This is a distinction between which datasets were considered (<500 yr resolution) and what intervals were analyzed (≥ 4 points in the analysis windows). We have clarified this throughout the manuscript.

Combining wet and dry excursions

I don't expect any centennial scale Holocene climate events to involve a change in significant planetary temperature, bar possibly the 8.2 ka event. And therefore by virtue of Clausius Clapeyron, a similar precipitation response. Therefore most, if not all, Holocene climate excursions will involve both warming and cooling, and wetting and drying in different places. The spatial analysis section of manuscript demonstrates this. Yet the time-series analysis separates the two (Figure 3), despite the authors commenting that combined wet and dry excursions provide for more significant climate excursions than the two alone for the 8.2ka excursion (line 236, line 257) and can be used to help characterise climate mechanisms related to the 4.8 ka excursion (line 278). I recommend up to 3 additional panels for figure 3 which combine wet and dry, hot and cold, and possibly all four would be useful to demonstrate this.

Thank you for this suggestion. We added a third panel to figure 3 to highlight joint warm/wet and cold/dry excursions, which helped clarify interaction between excursions in different parts of the climate system. We also have explored warm/cold and wet/dry joint excursions, but found them to be less helpful in understanding than by examining the separate responses already apparent in figure 3.

4.2, 4.0 and the recognition of contemporaneous events

The authors imply that Marchant & Hooghiemstra (2004) is an example of a compilation with poorer chronologies being tied to the 4.2ka event. Marchant & Hooghiemstra do not attempt to

link their compilation to the 4.2ka event. It is only the work of others later who have seized upon proximal changes and assigned 4.2 to these changes. I think not recognising this does a disservice to Marchant & Hooghiemstra and should be recognised in the text (paragraph lines: 146-169), otherwise a straw man is created.

Good point - we did not mean to imply that the study was originally intended to characterize the 4.2 ka event. We have revised that paragraph to clarify the origin of the compilation by Renssen.

This raises the issue of multiple climate excursions occurring with a few hundred years of each other – which must surely happen, and their potential ‘lumping’ by aggregators, which the authors allude to on line 354-357. The 4.0 and 4.4ka events can be different to the 4.2 ka event, and have been shown to be in the literature. Leipe et al., (Quaternary International, 2014) found both 4.4 and 4.0 events in pollen records from Tso Moriri. Scroxton et al. (2023, QSR 107837) found both 4.2 and 4.0 in different principal components of an Indian Ocean regional synthesis using a quantitatively determined dataset. Even the GSSP stalagmite for the Meghalayan, KM-A has two positive isotopic shifts at 4.3 and 4.1 ka (Berkelhammer et al., 2012).

The 4.0 tropical climate shift can be its own climate event (though not an excursion by the nomenclature used in this study), even without global signature or universal regional signature. The 200 year window used in this study may not be able to distinguish these events, whereas the 15 and 20 year windows of Scroxton et al. can. The paragraph between lines 351 and 362 could be updated to reflect separation of these two events in the literature, and the resolution limitations of a 400 year window.

Agreed. Multiple, legitimate excursions can occur in adjacent windows, or the same window, and our study design would obscure this. We have updated this section of the discussion to make this clear.

Globally Significant vs Globally Impacting Event

I think the manuscript also conflates globally impacting vs globally significant events. We should expect smaller climate excursions to decay with distance from their focus regions. Thus what is a significant event in one region may be comparable to other centennial scale variability in others, and an undetectable part of noise in others. Identification of a climate event in a particular area is important to characterise the spatial extent of a climate events, regardless of whether it has a magnitude greater than other centennial scale variability. The idea of excursions “smaller than those observed throughout the record” (line 412) is not a problem if you’re trying to determine the global impact of an event. To truly understand the 4.2 kyr event we need to know where the event impacted, as well as where it was significantly different to other climate variability. The methodology used here is addresses the latter. I think the authors should explicitly recognise that their analysis shows a limited spatial significance of the 4.2 ka event, not a limited spatial impact of the 4.2 ka event.

We agree that our study focuses on the spatiotemporal significance of the 4.2 ka event, and that the impact of a potential event is a different (although not entirely unrelated) question. We have

added text in multiple places in the manuscript, and added a paragraph at the end of the discussion that addresses this point, and highlights that many sites and some regions experienced events that could potentially have been very impactful. However, we believe that our results do also say something about the spatial impact of the event as well. The vast majority of datasets do not include excursions ca. 4.2 ka at all. It is not that there are events but they're typical for the Holocene - it is that there aren't events at all. So while we've made changes to the text to highlight our focus on significance, we also stand by the suggestion that the spatial impact of the event is limited, especially in comparison to claims that it is a global or hemispheric-scale event.

Minor Comments & Corrections

Line 35: "mitigating forced anthropogenic changes"

fixed

Line 93-95: This sentence needs rewriting to correct grammar. I doubt that your understanding has been ratified by the ICS.

fixed

Line 100: Sentence beginning "The 4.2 ka event" feels out of place. Could it be located either in the paragraph above, or below the 8.2 summary towards the bottom of the paragraph.

We modified this paragraph to improve this

Paragraph Line 126: This paragraph is a little hard to follow. I feel line 129 to 135 isn't necessary given what follows from Line 135 onwards. Key words and phrases from this section could be folded into what follows without losing any understanding.

We modified this paragraph to improve the clarity and flow

Line 184: Swap order? "only using the most significant excursion at each site for each time period".

We agree that this is confusing, and clarified by eliminating the second occurrence of "each dataset"

Fig 2b, c: y-axis is number not age?

Yes you are correct. We have corrected the axis labels in the revised manuscript.

Line 555: hyperlink is incorrect, link as spelt is correct

fixed

Fig S5: "four consecutive observations"?

fixed

Reviewer #1 (Remarks on code availability):

I have lightly assessed the code when checking other issues that came up during peer review. While I cannot vouch for its reusability, it does seem complete.

Reviewer #2 (Remarks to the Author):

This is a valuable addition to the literature and I recommend publication. It addresses a topic that pervades the literature—were there widespread globally significant changes in climate in the past? This topic has many adherents, and the literature is awash with claims of abrupt events in the Holocene, many of which lack scientific rigour and statistical significance. This paper uses a very large set of data and uses a rigorous statistical framework to address the topic. The results are particularly noteworthy in light of the International Geological Commission's decision to adopt the "Northgrippian" and "Meghalayan" (8.2ka and 4.2ka BP) as times of significant change in the Holocene. This paper affirms that the 8.2ka event was widespread, but clearly shows that there is no significance for the 4.2ka "event". It is thus an important rejoinder to the erroneous decision of the IGC. The authors do mention this point, but they could make more of it, I think. The IGC decision was simply wrong and should be reversed,

The paper clearly lays out its methods and data sources, allowing others to re-visit the topic if they wish. They also make the important point that, while there were certainly local or regional scale anomalies in the past, there were very few time periods when anomalies extended to a global or even hemispheric scale.

A recent paper by Ön, Z. Bora (in QSR, 2023) is not cited and is relevant as it conducts a similar statistical assessment of abrupt changes in climate in the Mediterranean/Middle East region, where the original idea of a "4.2ka event" originated (he also found little evidence for it). I think papers by Kathayat should be cited when discussing Mawmluh cave, as she collected more speleothems from the cave and found no support for Berkelhammer's claim for a 4.2ka event, (which foolishly became the type section designated by the IGC).

We thank the reviewer for their positive review. We have added discussion of both of these papers to the manuscript.

Reviewer #3 (Remarks to the Author):

Dear Editor

I have now completed my review of the McKay et al. manuscript. I found it very clear and well written and the treatment of the data robust (even if I'm not an expert of some methods). Therefore, conclusions appear supported by this extensive review. I'm therefore completely in favor of the final acceptance of this manuscript. This represent an important contribution in the discussion on Holocene climate variability and its "formal" division. The former is probably more important than the latter.

We thank the reviewer for the encouraging and constructive review. We address the questions, comments and concerns in detail below.

I have some general considerations/comments and some minor point along the text

1) I think there is not enough discussion on the fact that age models can contain a very large error and there would be a potential impact on the conclusions. So, records are selected also from this point of view or not? I think it would be a monumental effort to obtain all these data and treat them consequently, but maybe a figure showing the range of the error associated to the age models could be a significant step in the quality of the manuscript. I can agree that the large number of data used may reduce the impact.

We agree that age model uncertainty contributes greatly to the identification, and especially the evaluation of synchronicity of Holocene climate excursions. Rather than include age model uncertainty explicitly, which we agree would have been a monumental task that likely would not have shed any additional light on the problem, we 1) used relatively large (400 ± 100) yr windows to summarize the events, accommodating substantial (although not all) uncertainty. 2) Secondly we examined the prevalence of events in nearby intervals. If a synchronous excursion across a large region had occurred at 4.2 ka, we should expect this to be smoothed out and distributed to nearby bins in our analysis (i.e., 4.0 and 4.4 ka). Indeed, this is exactly what we see happening at the 8.2 ka event. However, we don't see elevated prevalence of excursions at 4.2 ka and nearby intervals, even though we expect age uncertainty to be lower during this more recent interval. We have added some text to 4.2 ka subsection in the discussion to clarify this.

2) It is obviously not fundamental, but data are mostly concentrated in the Northern Hemisphere (as noted in the manuscript), Can it bias some conclusions?

Yes, the geographic bias in the data is an important part of our dataset. In one sense, this does affect the results in that we cannot detect excursions where we don't have observations. However we were careful to spatially weight the data in our analysis, so the density of data in the northern Hemisphere does not disproportionately affect the results "global" results, nor the excursion maps.

3) It is clearly stated in the text that the interpretation of the proxies is based on the proposed interpretation in the original paper, it is also indicated that some proxies can be seasonally biased. However, it is not well specified as this has been considered.

Indeed, we kept track of the interpreted seasonality of the data, but did not explicitly analyze the data by season. In the literature, the seasonality of the 4.2 ka event is poorly defined, so we took a permissive approach, analyzing all seasonalities and using the most significant result for each time period at each site.

4) At the end seems that the 4.2 event is just one among many others during the Holocene, which has not enough spatial coherency to be considered a reliable boundary for the Middle to Late Holocene (even if the main conclusions are probably related to the meaning of climatic excursions during the Holocene). From the proposed analyses no other excursions seem to be suitable. Surprisingly (but not necessarily surprisingly) at 4.8 ka seems to occur some. Is it enough to say, searching for the 4.8 ka event? I think the author should try to suggest some line of research after this robust analyses. Or the main conclusion is: The Holocene has so very large temporal and regional variability that the boundary can be just an agreement?

Agreed. We have added the following sentence to the conclusion: "Furthermore, the prevalence of regional climate excursions in the Holocene, and the lack of global scale phenomena, suggests that climate excursions should not be used as Holocene GSSPs more generally."

Minor points

Why do you use the term "Medieval Warm Period" instead of "Medieval Climate Anomaly"? Is this because in your analyses "Medieval Period" is characterized by a clear temperature anomaly?

We agree that "Medieval Climate Anomaly" is the better choice here and have made this change.

Lines 65-67 May be is a matter of room. But here some references would be necessary.

Agreed. We added references to this discussion

Figure 1: Fig. 1a, not clear the caption. There are three color in the figure, but it is unclear what it is the dark brown.

That's the overlap between the two histograms, which we have clarified in the caption

Figure 2 Maybe the caption of the figure 2 can be improved, for instance the color in figure b & c could not be obvious without an explanation.

We agree, and have improved the caption

Lines 259-261 I'm wondering if it is just related to the fact that many authors searched for cold and dry climate.....

It's possible there are biases in the literature, although across the breadth of the Holocene, a lot of authors have likely searched for a lot of different types of phenomena, at many different times. It's impossible to know for sure, however, because our study focuses on all of the data from records and not events, we think it's likely this effect doesn't affect our conclusions.

Lines 343-346 here it would be correct to quote Bini et al. 2019. Despite the analyses of Bini et al was very qualitative, they show the very complex nature of the signal at regional scale.

Agreed, we've added a citation to this paper here (and elsewhere).

Lines 351-354 Hmm....be honest this is not a trivial point. Of course this can be the same for many other "events" in the Holocene. But if we align all the record centered at 4.2 ka, is this event become significant? I mean, the discussion of the error is not trivial. The authors give an answer in the following sentences but probably not necessarily all readers would be convinced.

This is essentially the question we explore in Fig S3 - if we consider both time periods together can we get more of a "classic" 4.2 ka response. We do see more of what is described in the literature, but the expression is still quite regionally restricted. We also added further discussion of the impact of chronological uncertainty here.

When I was younger in the region where I worked there were very good evidences that the 4.2 event was dry and well chronologically constrained. Moving around I have discovered that it is not the case. I think part is related to the ability of the archives to record events, partly depending on the proxy you use.....but at the end McKay et al can have right and 4.2 event is not so prominent. One among other complex climatic configuration during the Holocene. So, my indication is that this manuscript can be accepted after minor/moderate modifications. Probably some more suggestions on future work need to be inserted in the conclusions and in particular the main effort in disentangling this complex climatic variability, since model, at best, can give us just a simplified vision of the future.

Thank you for this comment and your review - in response this, and reviewer 1, we added further discussion of how impactful the event can be locally or regionally, even if it's not remarkable in the context of the Holocene.

Reviewer #3 (Remarks on code availability):

ok

REVIEWERS' COMMENTS

Reviewer #1 (Remarks to the Author):

An excellent revised manuscript that is ready for publication. It is timely, relevant, of broad interest, and has important results achieved through novel methodology. The authors have responded well to my concerns and those of the other reviewers, addressing most issues, and adequately justifying where they feel no changes were needed for others. I agree with the overall point made by the authors in the rebuttal that a broad criteria approach is necessary. It is clear that the 4.2 ka event needs analysing at both broad (low false negatives) and fine (low false positives) scale. Given that there are other manuscripts in production attempting the latter, this approach is an important part of the jigsaw in understanding abrupt climate events in the Holocene, with good discussion on the wider ramifications.

Minor comment: In the methodology, some papers are referred to mid-sentence by number (Line 486: 14, Line 488: Ön, Line 494: 13, Line 505, 13), and some by name., Consistency should be applied here.

Reviewer #3 (Remarks to the Author):

I think the authors aswered to all my comments. It is very welcome manuscript. For me is acceptable.

Reviewer #3 (Remarks on code availability):

No other suggestions

Response to reviewer comments.

We thank the reviewers for their encouraging reviews and for recommending the paper for acceptance. Please find our responses and comments on specific sections below. Reviewer comments are in black, and our responses are in blue.

REVIEWERS' COMMENTS

Reviewer #1 (Remarks to the Author):

An excellent revised manuscript that is ready for publication. It is timely, relevant, of broad interest, and has important results achieved through novel methodology. The authors have responded well to my concerns and those of the other reviewers, addressing most issues, and adequately justifying where they feel no changes were needed for others. I agree with the overall point made by the authors in the rebuttal that a broad criteria approach is necessary. It is clear that the 4.2 ka event needs analysing at both broad (low false negatives) and fine (low false positives) scale. Given that there are other manuscripts in production attempting the latter, this approach is an important part of the jigsaw in understanding abrupt climate events in the Holocene, with good discussion on the wider ramifications.

Thank you for these comments and for your helpful reviews throughout the process.

Minor comment: In the methodology, some papers are referred to mid-sentence by number (Line 486: 14, Line 488: Ön, Line 494: 13, Line 505, 13), and some by name., Consistency should be applied here.

We've standardize this to include names throughout.

Reviewer #3 (Remarks to the Author):

I think the authors aswered to all my comments. It is very welcome manuscript. For me is acceptable.

Reviewer #3 (Remarks on code availability):

No other suggestions

Thank you for these encouraging words and your review